# Safe Multi-agent Reinforcement Learning with Protection Motivation Theory

## Abstract

A challenging problem for implementing multi-agent reinforcement learning (MARL) in real-world applications is ensuring the safety of cooperative strategies. According to the Protection Motivation Theory (PMT), threat appraisals result in negative emotions and elicit protective behaviors, which are instrumental for coping with security threats. Drawing inspiration from the PMT, we focus on two discrete emotions–fear and regret–to evaluate threat severity and facilitate multiple agents to learn protective behaviors. These can promote cooperative decision-making with fewer safety violations. Specifically, we propose two safety guarantee methods with PMT: fear for safety guarantee (F4SG) and regret for safety guarantee (R4SG), utilizing the active inference technique to model the emotions of fear and regret separately. The threat severity evaluated by these emotions influences the state value and the executed action respectively, which avoids the potential threat of visiting certain states or taking certain actions. Experimental results demonstrate that our proposed methods are safer and more efficient than state-of-the-art baselines on challenging tasks in safe MARL benchmarks.

## 1 Introduction

In recent years, Multi-agent Reinforcement learning (MARL) has become an active research field with great success in addressing various complex multi-agent decision-making tasks, such as multi-robot control (Chen et al., 2024), autonomous vehicle coordination (Li et al., 2024), and transportation resource management (Zhang et al., 2023). Although traditional MARL methods show significant performance for completing cooperative tasks, the lack of safety guarantees limits their deployment in the real world. These methods are purely for reward maximization, completely ignoring safety violations (Gu et al., 2023). For example, the autonomous vehicles controlled by MARL agents ignore the risk of collision with others in order to obtaining high returns through high-speed driving; the robots controlled by MARL agents ignore irreversible harm to themselves or environmental elements in order to quickly completing the required tasks.

Recently, some works offer effective learning algorithms for ensuring safety in MARL methods: 1) The primal-dual framework-based safe MARL methods (Lu et al., 2021; Ding et al., 2023; Gu et al., 2023). These methods formulate the constrained optimization problem in safe MARL as the min-max problem between reward and cost, and utilize a primal-dual framework (Boyd et al., 2004) or its combination with trust region approach (Schulman et al., 2015) to find the saddle point. 2) The shielding-based safe MARL methods (Elsayed-Aly et al., 2021; Melcer et al., 2022; Xiao et al., 2023; Melcer et al., 2024). These methods extend the shielding framework developed in the single-agent setting (Alshiekh et al., 2018) to the multi-agent setting, preventing multiple agents from exploring any unsafe states or actions that violate the safety specification expressed by linear temporal logic (Pnueli, 1977). 3) The safe MARL methods with safety layer (Sheebaelhamd et al., 2021; Shi et al., 2023). These methods add the data-driven safety layer for mapping unsafe actions to safe actions, enhancing the capability of correcting unsafe behaviors. 4) Other approach for safe MARL methods (Zhu et al., 2020; Jusup et al., 2024). These methods introduce the novel prior

knowledge to ensure the safety, such as gaussian processes and mean field. However, the above works ignore the great potential of utilizing human knowledge about coping with threats in guiding MARL methods to ensure safety, which can effectively achieve the goal of improving rewards while satisfying safety constraints.

The Protection Motivation Theory (PMT (Rogers, 1975)) originated from social psychology suggests that threat appraisals result in the emotion of fear and impact protective behaviors. PMT explains human security behaviors to protect themselves when perceiving threats. (Ogbanufe & Pavur, 2022) further extends PMT framework and analyzes the effects of discrete emotions (i.e. fear and regret) for security protection behaviors. Naturally, PMT is also applicable to guide MARL methods in learning safe cooperative strategies in safety-critical scenarios: agents actively infer discrete emotions to evaluate the threat severity and influence the current cooperative decision-making, avoiding the potential threat of visiting certain states or taking certain actions.

In this paper, we introduce PMT into the MARL to address the challenge safety. With PMT, we propose two novel safe guarantee methods: fear for safety guarantee (F4SG) and regret for safety guarantee (R4SG). Same as the cognitive mediational process (threat appraisal and coping appraisal) in PMT, each of our methods takes two steps to implement the security protection behaviors elicited by discrete emotions. First, we utilize the active inference technique to model the emotions of fear and regret separately, mapping the state to the severity of fear and the state-action to the severity of regret. Second, the state value and the executed action are respectively influenced according to the threat appraisal, facilitating the learning of security protection behaviors. Besides, we take Lagrange dual framework combined with trust region approach to formulate the constrained optimisation problem as the min-max problem, accelerating the learning process of safety cooperative strategy.

To summarize, our main contributions are three-fold: 1) We provide a novel perspective, with the Protection Motivation Theory originated from social psychology, for safe multi-agent reinforcement learning, introducing the human knowledge about facing threats into MARL to ensure safety. 2) With PMT, we propose two effective safety guarantee methods–fear for safety guarantee (F4SG) and regret for safety guarantee (R4SG)–to facilitate multiple agents learning safety protective cooperative behaviors. 3) Experimental results show that the two safety guarantee methods outperform the state-of-the-art baselines in terms of balancing performance and constraint satisfaction in several challenging tasks on three safe MARL benchmarks.

## 2 Preliminaries

### 2.1 Problem Formulation

The safe MARL problem can be formulated as a constrained Markov decision process (CMDP) (Gu et al., 2023). Considering a fully-cooperative setting, CMDP is described by a tuple $< \mathcal{N}, \mathcal{S}, \mathcal{A}, \mathcal{T}, \mathcal{R}, \mathcal{C}, \boldsymbol{c}, \gamma, \rho^0 >$, where $\mathcal{N}$ is the set of $n$ agents indexed by $1, 2, \ldots, n$; $\mathcal{S}$ is the state space shared for all agents; $\mathcal{A} = A^1 \times A^2 \times \ldots \times A^n$ is the joint action space and $A^i$ denotes $i$th agent's action space; $\mathcal{T} : \mathcal{S} \times \mathcal{A} \to \mathcal{S}$ is the probabilistic transition function, and at each time step $t$, the agents in state $s_t \in \mathcal{S}$ select actions $\mathbf{a}_t \in \mathcal{A}$ according to their policy $\pi^i(\mathbf{a}^i|s_t)$, then reach the new state $s_{t+1} \sim \mathcal{T}(s_t, \mathbf{a}_t)$; $\mathcal{R} : \mathcal{S} \times \mathcal{A} \to \mathbb{R}^n$ is the reward function shared for all agents; $\mathcal{C} = \{\mathcal{C}_k^i\}_{1 \le k \le m^i}^{i \in \mathcal{N}}$ is the set for the sets of cost functions (every agent $i$ has $m^i$ cost functions) with the form $\mathcal{C}_k^i : \mathcal{S} \times A^i \to \mathbb{R}$; $\boldsymbol{c} = \{c_k^i\}_{1 \le k \le m^i}^{i \in \mathcal{N}}$ is the set of corresponding cost-constraining values; and $\gamma \in [0, 1)$ is the discount factor. It aims to find a joint policy $\boldsymbol{\pi} = \prod_{i=1}^n \pi^i$, making all agents work cooperatively and safely, by solving the following constrained optimization problem:

$$\mathcal{J}(\boldsymbol{\pi}) \triangleq \mathbb{E}_{s \sim \mathcal{T}, \mathbf{a} \sim \boldsymbol{\pi}}[\sum_{t=0}^{\infty} \gamma^t \mathcal{R}(s_t, \mathbf{a}_t)],$$

$$s.t. \ \mathcal{J}_k^i(\boldsymbol{\pi}) \triangleq \mathbb{E}_{s \sim \mathcal{T}, \mathbf{a} \sim \boldsymbol{\pi}}[\sum_{t=0}^{\infty} \gamma^t \mathcal{C}_k^i(s_t, \mathbf{a}_t^i)] \le c_k^i, \ \forall i = 1, 2, \ldots, n; \ k = 1, 2, \ldots, m^i.$$

(1)

## 2.2 Protection Motivation Theory

PMT (Rogers, 1975) is a long-studied and persuasive theory in social psychology, which suggests that threat appraisals result in the emotion of fear and individuals will respond to threat appraisals via protective behavior. Thus, emotions are instrumental for coping with security threats (Liang et al., 2019; Wang et al., 2023). PMT consists of three parts: information sources, cognitive mediational process, and coping patterns. The information source refers to an individual's cognition of various factors in the environment; The cognitive mediational process is the core part of this theory, which includes two elements: threat assessment and response assessment; Coping patterns refer to the behavior of individuals after undergoing the cognitive mediational process. This theory explains the reasons why individuals change their behavior due to environmental and social factors through cognitive mediational, and comprehensively elaborates on the process and mechanism of cognition to behavior change. PMT has a wide range of applications, such as information systems, health, and marketing. With the evolution of PMT, many studies consider various emotions affecting protective behavior to explain individuals' protection motivation (Leary, 2007; Cho et al., 2020; Nehme & George, 2022; Ogbanufe & Pavur, 2022). According to the theoretical underpinning of recently revisited PMT (Ogbanufe & Pavur, 2022), the behaviors that characterize individuals' motivation to protect themselves from a threat are elicited by discrete emotions of fear and regret, which have different effects on security protection behaviors. Naturally, PMT is also applicable to multi-agent systems in safety-critical scenarios and we facilitate multiple agents learning safety cooperative strategy with this theory.

## 2.3 Active Inference Technique

Active inference technique (Friston et al., 2009) offers a useful computational framework for minimizing the divergence between an unbiased model of the real world and a biased generative model of organisms' preferences, which arises from the free energy principle (Friston, 2010). Given a generative model, it tends the output distribution $q(x; \theta)$ (known as the prior in Bayesian) towards the true posterior $p(x|y)$ via minimizing the free energy $\mathcal{F}$:

$$\mathcal{F} = D_{\mathrm{KL}}[q(x;\theta)||p(x,y)] \tag{2}$$

where $D_{\mathrm{KL}}(\cdot||\cdot)$ is the Kullback-Leibler (KL) divergence. $x$ denotes the hypotheses about the world and $y$ denotes the observation about the world. The model parameters $\theta$ are treated as random variables, casting the learning of the model as a process of inference.

# 3 Safe Multi-agent Reinforcement Learning with Protection Motivation Theory

This section describes our proposed safety guarantee methods with protection motivation theory, F4SG and R4SG, that elicit agents' protective behaviors via discrete emotions of fear and regret. We follow the Centralized Training and Decentralized Execution framework, where agents are trained with access to global information but make decisions based on their own local information in execution. Both F4SG and R4SG follow the cognitive mediational process revealed by PMT. We then introduce the details of implementing our safety guarantee algorithm on the representative Actor-Critic style MARL methods.

## 3.1 Fear for Safety Guarantee

### 3.1.1 Threat Appraisal about Fear

Fear increases an individual's perception of threats and prompts them to take suggested actions to reduce the fear or harm caused by such threats. The fear-based PMT suggests that fear acting as a mediator between threat appraisals and protection motivation can influence individuals' safety protection behaviors to avoid potential threats. The observations in neurosciences (Herry et al., 2008; Jovasevic et al., 2015) suggest that the fear-inducing underlying mechanism is state dependent. Thus, we propose a fear prior network (FPN) learned via active inference that maps agents' states to their fear severity.

For the emotion of fear, it can be elicited by certain negative stimuli (LeDoux & Daw, 2018). In safe MARL, the cost function serves as a negative incentive used to punish the unsafe behaviors of the agent, which can be viewed as a form of negative stimulus. Furthermore, the emotion of fear can also be caused by uncertainties in individual perception (Carleton & Nicholas, 2016). Therefore, the FPN considers both the anticipated negative stimuli and epistemic uncertainties simultaneously. Mathematically, the fear severity inferred by FPN for each agent $i$ can be expressed as $\tilde{f}^i = FPN(s; \zeta^i)$. Formally, the fear severity of state $s$ is the probability value at the $1st$-index of $\tilde{f}^i$, while the probability value at the other index denotes the level of non-fear.

To make the inferred fear severity $\tilde{f}^i$ about state $s$ incorporate both the elicited factors, the FPN is optimized by minimizing the following:

$$\text{argmin} \left( D_{\text{KL}}[q(\tilde{f}^i)||p(\mathcal{C}^i)] - ln[p(\mathcal{C}^i)] \right) + D_{\text{KL}}[FPN(s), FPN(s_{\text{D}})] \tag{3}$$

where the first term is the free energy simplified by Variational Bayes (Buckley et al., 2017) and the second term is the epistemic uncertainty between the fear severity of state current state $s$ and the fear severity of state history state $s_{\text{D}}$ (stored in replay buffer D); $p(\cdot)$ and $q(\cdot)$ are the probability distributions; $D_{\text{KL}}$ is the Kullback-Leibler (KL) divergence. Due to the true distribution of cost is binomial distribution, we adopt the reparameterization trick technique (Jang et al., 2017) to sample from $q(\tilde{f}^i)$ for generating the binomial distribution about fear severity $b(\tilde{f}^i)$. Meanwhile, the second term can be expressed as the cognitive error, which is replaced by the mean squared loss between $\tilde{f}^i$ and $\tilde{f}_{\text{D}}^i$ in the implementation. Therefore, the final loss function to learn the FPN is:

$$\mathcal{L}(\zeta) = D_{\text{KL}}[b(\tilde{f}^i; \zeta)||p(\mathcal{C}^i)] + L_2(\tilde{f}^i, \tilde{f}_{\text{D}}^i; \zeta) \tag{4}$$

### 3.1.2 Coping Response to Fear

After modeling the threat severity of fear $\tilde{f}^i$ at state $s$, it seeks to ensure the safety of MARL methods. The safe MARL problem is formulated as the following constrained optimization problem:

$$\max \ \mathbb{E}_{s \sim \mathcal{T}_{\boldsymbol{\pi}_\theta}, \mathbf{a} \sim \boldsymbol{\pi}_\theta}[A_{\boldsymbol{\pi}_\theta}(s, \mathbf{a})]$$
$$\text{s.t. } \mathbb{E}_{s \sim \mathcal{T}_{\boldsymbol{\pi}_\theta}, \mathbf{a} \sim \boldsymbol{\pi}_\theta}[A_{k, \boldsymbol{\pi}_\theta}^i(s, \mathbf{a})] \leq c_k^i, \ i = \{1, 2, \ldots, n\}; \ k = \{1, 2, \ldots, m^i\} \tag{5}$$

where $A_{\boldsymbol{\pi}_\theta}(s, \mathbf{a})$ denotes the advantage estimate relying on rewards and $A_{k, \boldsymbol{\pi}_\theta}^i(s, \mathbf{a})$ denotes the cost advantage estimate relying on costs. According to Lagrange duality theory, the Lagrange dual problem about the Eq.(5) can be derived as:

$$\max_\theta \min_\lambda \left[ \mathbb{E}_{s \sim \mathcal{T}_{\boldsymbol{\pi}_\theta}, \mathbf{a} \sim \boldsymbol{\pi}_\theta}[A_{\boldsymbol{\pi}_\theta}(s, \mathbf{a})] - \sum_{i=1}^{n} \sum_{k=1}^{m^i} \lambda_k (\mathbb{E}_{s \sim \mathcal{T}_{\boldsymbol{\pi}_\theta}, \mathbf{a} \sim \boldsymbol{\pi}_\theta}[A_{k, \boldsymbol{\pi}_\theta}^i(s, \mathbf{a})] - c_k^i)] \right] \tag{6}$$

where $\lambda$ denotes the dual variable, and $\lambda \geq 0$. To guarantee satisfying constraints during training theoretically and make the stable training process (Kim et al., 2023; Gu et al., 2023), the trust region approach is applied to the above min-max optimization problem. So for agent $i_h \in i_{1:h}$ ($i_{1:h}$ denotes any subset of $\mathcal{N}$), Eq.(6) is written as:

$$\max_{\theta^{i_h}} \min_{\lambda_{1:m^{i_h}}^{i_h}} \left[ \mathbb{E}_{s \sim \mathcal{T}_{\boldsymbol{\pi}_{\theta_{t_s}}}, \mathbf{a}^{i_{1:h-1}} \sim \boldsymbol{\pi}_{\theta_{t_s+1}^{i_{1:h-1}}}^{i_{1:h-1}}, \mathbf{a}^{i_h} \sim \pi_{\theta^{i_h}}^{i_h}} [A_{\boldsymbol{\pi}_{\theta_{t_s}}}^{i_h}(s, \mathbf{a}^{i_{1:h-1}}, \mathbf{a}^{i_h})] \right.$$
$$\left. - \sum_{k=1}^{m^{i_h}} \lambda_k^{i_h} (\mathbb{E}_{s \sim \mathcal{T}_{\boldsymbol{\pi}_{\theta_{t_s}}}, \mathbf{a}^{i_h} \sim \pi_{\theta^{i_h}}^{i_h}} [A_{k, \boldsymbol{\pi}_{\theta_{t_s}}}^{i_h}(s, \mathbf{a}^{i_h})] + J_k^{i_h}(\boldsymbol{\pi}_{\theta_{t_s}}) - c_k^{i_h})] \tag{7}$$

meanwhile satisfying $\bar{D}_{\text{KL}}(\pi_{\theta_{t_s}^{i_h}}^{i_h}, \pi_{\theta^{i_h}}^{i_h}) \triangleq \mathbb{E}_{s \sim \mathcal{T}_{\boldsymbol{\pi}_{t_s}}}[D_{\text{KL}}(\pi_{\theta_{t_s}^{i_h}}^{i_h}(\cdot|s), \pi_{\theta^{i_h}}^{i_h}(\cdot|s))] \leq \delta$, which is processed by clip operator (Yu et al., 2022) in the implement. $J_k^{i_h}(\boldsymbol{\pi}_{\theta_{t_s}})$ is the cost objective function about policy $\boldsymbol{\pi}_{\theta_{t_s}}$, which guarantees that policy that ensures monotonic improvement property also satisfies safety constraints.

The fear severity from the fear prior network is applied to calculate cost advantage estimates in the second term. Specifically, the cost generalized advantage estimate is written as in its definition (Schulman et al., 2016) and expanded in a telescoping sum:

$$\hat{A}^{i_h}_{k,\boldsymbol{\pi}_{\theta_{t_s}}}(s_{t_s},\mathbf{a}^{i_h}_{t_s}) = \lim_{L\to\infty}\sum_{l=0}^{L}\gamma^l[Q^{i_h}_{k,\boldsymbol{\pi}_{\theta_{t_s+l}}}(s_{t_s+l},\mathbf{a}^{i_h}_{t_s+l}) - V^{i_h}_{k,\boldsymbol{\pi}_{\theta_{t_s+l}}}(s_{t_s+l})]$$

$$= \lim_{L\to\infty}\sum_{l=0}^{L}\gamma^l \mathcal{C}^{i_h}_k(s_{t_s+l},\mathbf{a}^k_{\boldsymbol{\pi}_{\theta_{t_s+l}}}) - V^{i_h}_{k,\boldsymbol{\pi}_{\theta_{t_s}}}(s_{t_s}) + \gamma^L V^{i_h}_{k,\boldsymbol{\pi}_{\theta_{t_s}}}(s_{t_s+l}) \quad (8)$$

$$= \sum_{l=0}^{\infty}\gamma^l \mathcal{C}^{i_h}_k(s_{t_s+l},\mathbf{a}^k_{\boldsymbol{\pi}_{\theta_{t_s+l}}}) - V^{i_h}_{k,\boldsymbol{\pi}_{\theta_{t_s}}}(s_{t_s})$$

Eq.(8) reveals the importance of the cost function. Fear severity $\tilde{f}^{i_h}_k \in [0,1]$ replaces the cost function $\mathcal{C}^{i_h}_k \in \{0,1\}$ to improve the accuracy and smoothness of cost advantage estimation at state $s_{t_s}$. The cost advantage estimation relying on fear severity is:

$$\hat{A}^{i_h,(F)}_{k,\boldsymbol{\pi}_{\theta_{t_s}}}(s_{t_s},\mathbf{a}^{i_h}_{t_s}) = \sum_{l=0}^{\infty}\gamma^l \tilde{f}^{i_h}_k(s_{t_s+l}) - V^{i_h}_{k,\boldsymbol{\pi}_{\theta_{t_s}}}(s_{t_s}) \quad (9)$$

Then, we introduce the implemetation process of F4SG. For additional information, please refer to A.4. The actor networks is trained to minimize the Eq.(10):

$$\mathcal{L}^{i_h}_{(\lambda)}(\theta^{i_h}) = \frac{1}{B}\sum_{b=1}^{B}\sum_{t=0}^{T_s}\min[\frac{\pi^{i_h}_{\theta^{i_h}}(a^{i_h}_i|o^{i_h}_i)}{\pi^{i_h}_{\theta^{i_h}_{\hat{t}}}(a^{i_h}_i|o^{i_h}_i)}L^{\theta^{i_h}}_{\lambda_k}, \text{clip}(\frac{\pi^{i_h}_{\theta^{i_h}}(a^{i_h}_i|o^{i_h}_i)}{\pi^{i_h}_{\theta^{i_h}_{\hat{t}}}(a^{i_h}_i|o^{i_h}_i)},1\pm\epsilon)L^{\theta^{i_h}}_{\lambda_k}],$$

$$\text{where } L^{\theta^{i_h}}_{\lambda_k} = \hat{A}^{i_h}(s_t,\mathbf{a}_t) - \sum_{k=1}^{m^{i_h}}\lambda^{i_h}_k \hat{A}^{i_h,(F)}_k(s_t,\mathbf{a}^{i_h}_t). \quad (10)$$

The dual variable $\lambda$ is calculated as Eq.(11):

$$\lambda^{i_h}_k \leftarrow [\lambda^{i_h}_k - \alpha_\lambda \frac{-1}{B}\sum_{b=1}^{B}(S^{i_h}_{J^{i_h}_k,c^{i_h}_k} + I^{i_h}_k)]_+$$

$$\text{where } S^{i_h}_{J^{i_h}_k,c^{i_h}_k} = (1-\gamma)\frac{1}{BT_s}\sum_{b=1}^{B}\sum_{t=1}^{T_s}\hat{V}^{i_h}_k(s_t) - c^{i_h}_k, I^{i_h}_k = \sum_{t=0}^{T_s}\frac{\pi^{i_h}_{\theta^{i_h}}(a^{i_h}_i|o^{i_h}_i)}{\pi^{i_h}_{\theta^{i_h}_{\hat{t}}}(a^{i_h}_i|o^{i_h}_i)}\hat{A}^{i_h,(F)}_k(s_t,\mathbf{a}^{i_h}_t) \quad (11)$$

The critic networks and cost networks are trained to minimize the Eq.(12a) and Eq.(12b) respectively:

$$\mathcal{L}^C(\phi) = \frac{1}{BT_s}\sum_{b=1}^{B}\sum_{t=0}^{T_s}max([V_\phi(s_t) - \hat{R}_t]^2, [\text{clip}(V_\phi(s_t),V_{\phi_{\hat{t}}}(s_t)\pm\epsilon) - \hat{R}_t]^2) \quad (12a)$$

$$\mathcal{L}^C_k(\phi) = \frac{1}{BT_s}\sum_{b=1}^{B}\sum_{t=0}^{T_s}max([V_{k,\phi}(s_t) - \hat{\mathcal{C}}_t]^2, [\text{clip}(V_{k,\phi}(s_t),V_{k,\phi_{\hat{t}}}(s_t)\pm\epsilon) - \hat{\mathcal{C}}_{k,t}]^2) \quad (12b)$$

### 3.2 Regret for Safety Guarantee

#### 3.2.1 Threat Appraisal about Regret

Regret is a negative emotion of not performing a behavior that would positively contribute to one's personal goals. The regret-based PMT suggests that regret is important in individuals' decision-making to avoid the actual experience of meeting threats via the regret in pre-behaviors. Many works (Bastani et al., 2022; Karpov & Zhang, 2024) utilize the regret theory to facilitate the learning of policy. However, these works learn no-regret policies maximizing rewards by regret theory but ignore the great potential of regret in ensuring

safety. Intuitively, the regret severity for organisms is closely related to their current state and executed action. Thus, we propose a regret prior network (RPN) learned via active inference that maps the states and actions of agents to their regret severity.

Similar to the emotion of fear, the emotion of regret is elicited by two factors: certain negative stimuli (LeDoux & Daw, 2018) and gaps between actual and expectation (Liao et al., 2016). Consequently, the regret severity of state $s$ and action available actions $\mathsf{a}^i$ can be expressed as $\bar{r}^i = RPN(S, \mathsf{a}^i; \psi)$, where $\bar{r}^i$ has the same dimension $|A^i|$ with available actions. It represents the regret sever of each action in agent's available actions set at current state. Research (Kuhnle & Sinclair, 2011) suggests that regretting at every moment is harmful to humans, so we focus on the state-action violating safety. The regret severity $\bar{r}^i$ is processed by combining with the cost $\mathcal{C}$ of environment feedback:

$$\tilde{r}^i = \begin{cases} \bar{r} \ , & \mathcal{C}(s, \mathsf{a}^i) = 1 \\ \mathbf{0}^{|A^i|}, & \mathcal{C}(s, \mathsf{a}^i) = 0 \end{cases} \tag{13}$$

To make the inferred regret severity $\tilde{r}^i$ about state $s$ and available actions $\mathsf{a}^i$ incorporate both the elicited factors, the RPN is optimized by minimizing the following:

$$\mathcal{L}(\psi) = D_{\mathrm{KL}}[q(\tilde{r}^i; \psi)||\mathbb{I}(\mathcal{C}(s, \mathsf{a}^i) = 1) \cdot p(\mathsf{a}^i)] + L_2(\tilde{r}^i, \tilde{r}_{\mathrm{D}}^i; \psi) \tag{14}$$

where $\mathbb{I}(\cdot)$ is an indicator function and $p(\mathsf{a}^i)$ denotes the probability distributions of agent $i$'s actions under the policy $\boldsymbol{\pi}_{\theta^i}$.

### 3.2.2 Coping Response to Regret

For the safe MARL problem, the R4SG approach also optimizes the formulation expressed as Eq.(7), and most symbols explained in the F4SG approach are also applied to the R4SG approach. Together, the different symbols in the R4SG approach are explained and illustrated in the following. To maintain the optimal policy invariance, the regret severity does not directly influence the executed actions. It forms the security inductive bias to promote the actor network to learn the safe cooperative policy. For additional information, please refer to A.5. Specifically, the policy with security inductive bias in the trust region-based Lagrange dual problem is learned by minimizing the following loss function:

$$\mathcal{L}_{(\lambda)}^{i_h}(\theta^{i_h}) = \frac{1}{B} \sum_{b=1}^{B} \sum_{t=0}^{T_s} \min[\frac{(\mathbf{1} - \Omega_\omega(\tilde{r}_{\theta^{i_h}}^{i_h}))\pi_{\theta^{i_h}}^{i_h}(a_i^{i_h}|o_i^{i_h})}{(\mathbf{1} - \Omega_\omega(\tilde{r}_{\theta_{\hat{t}}^{i_h}}^{i_h}))\pi_{\theta_{\hat{t}}^{i_h}}^{i_h}(a_i^{i_h}|o_i^{i_h})} L_{\lambda_k}^{\theta^{i_h}},$$

$$\mathrm{clip}(\frac{(\mathbf{1} - \Omega_\omega(\tilde{r}_{\theta^{i_h}}^{i_h}))\pi_{\theta^{i_h}}^{i_h}(a_i^{i_h}|o_i^{i_h})}{(\mathbf{1} - \Omega_\omega(\tilde{r}_{\theta_{\hat{t}}^{i_h}}^{i_h}))\pi_{\theta_{\hat{t}}^{i_h}}^{i_h}(a_i^{i_h}|o_i^{i_h})}, 1 \pm \epsilon) L_{\lambda_k}^{\theta^{i_h}}], \tag{15}$$

$$\text{where } L_{\lambda_k}^{\theta^{i_h}} = \hat{A}^{i_h}(s_t, \mathbf{a}_t) - \sum_{k=1}^{m^{i_h}} \lambda_k^{i_h} \hat{A}_k^{i_h}(s_t, \mathbf{a}_t^{i_h}).$$

where $\Omega_\omega(\cdot)$ denotes the mask operation based on hyperparameter $\omega$, which follows the fact that human only regret a finite actions among all available actions and avoids the long tail distribution problem caused by the large action space. The dual variable $\lambda$ is similar as Eq.(11), which replaces $I_k^{i_h}$ with Eq.(16):

$$\tilde{I}_k^{i_h} = \sum_{t=0}^{T_s} \frac{(\mathbf{1} - \tilde{r}_{\theta^{i_h}}^{i_h})\pi_{\theta^{i_h}}^{i_h}(a_i^{i_h}|o_i^{i_h})}{(\mathbf{1} - \tilde{r}_{\theta_{\hat{t}}^{i_h}}^{i_h})\pi_{\theta_{\hat{t}}^{i_h}}^{i_h}(a_i^{i_h}|o_i^{i_h})} \hat{A}_k^{i_h}(s_t, \mathsf{a}_t^{i_h}) \tag{16}$$

where $\hat{A}_{k, \boldsymbol{\pi}_{\theta_{t_s}}}^{i_h}(s_{t_s}, \mathsf{a}_{t_s}^{i_h})$ is expressed in Eq.(8). The new dual variable $\lambda$ is calculated as Eq.(17):

$$\lambda_k^{i_h} \leftarrow [\lambda_k^{i_h} - \alpha_\lambda \frac{-1}{B} \sum_{b=1}^{B} (S_{J_k^{i_h}, c_k^{i_h}}^{i_h} + \tilde{I}_k^{i_h})]_+ \tag{17}$$

## 4    Experiments

To demonstrate that the agents implemented via F4SG/R4SG can cooperatively learn to achieve high rewards while satisfying their safety constraints, we evaluate F4SG and R4SG in three complex cooperative multi-agent environments with safety-critical tasks (Gu et al., 2023): Safe Multi-Agent MuJoCo (Gu et al., 2023), Safe Multi-Agent Isaac Gym (Gu et al., 2023), and Multi-Agent Power Distribution Networks (Wang et al., 2021), as shown in Fig. A.2. All methods are trained on a Ubuntu 20.04 operation system with CPU Intel Xeon E5-2630 v3 and GPU Tesla P40 and the neural network framework is built on Pytorch (Paszke et al., 2019). All presented results are average performance over 5 random seeds. The shaded area in each figure is the standard deviation.

### 4.1    Baselines

F4SG and R4SG are compared with various types of state-of-the-art MARL methods used as baselines:

- *Vanilla MARL*. We employ Multi-agent Proximal Policy Optimization (MAPPO (Yu et al., 2022)) and Heterogeneous-agent Proximal Policy Optimization (HAPPO (Kuba et al., 2022)) as two vanilla MARL baselines, representing the classical Actor-Critic style MARL method and the trust region Actor-Critic style MARL method.

- *Safe MARL*. Multi-agent Constrained Policy Optimisation (MACPO (Gu et al., 2023)) and MAPPO-Lagrangian (MAPPO-L (Gu et al., 2023)) are adopted as two safe MARL baselines.

### 4.2    Safe Multi-Agent MuJoCo Environment

#### 4.2.1    Experimental Settings

Safe MAMuJoCo, an extension of MAMuJoCo (Peng et al., 2021), retains in terms of background environment, agents, physics simulator, and the reward function. It adds the obstacles (i.e. walls) and the cost functions from (Zanger et al., 2021). For this environment, it requires $n$-agent Ant walking though corridor cooperatively and safely. The width of the corridor set by two walls is 10m and $n$-agent Ant receives a cost feedback if the distance between the Ant and the wall is less than 1.8m or the Ant topples over, expressed as:

$$c_t = \begin{cases} 0, \text{ for } 0.2 \leq \mathbf{z}_{\text{torso},t+1} \leq 1.0 \text{ and } z_{rot} > -0.7 \\ \qquad \text{and } ||\mathbf{x}_{\text{torso},t+1} - \mathbf{x}_{\text{wall}}||_2 \geq 1.8, \\ 1, \text{ otherwise.} \end{cases}$$

where $\mathbf{z}_{\text{torso},t+1}$ is the ant's torso's z-coordinate, $z_{rot}$ is the ant's rotation's z-coordinate, $\mathbf{x}_{\text{torso},t+1}$ is the ant's torso's x-coordinate, and $\mathbf{x}_{\text{wall}}$ is the x-coordinate of the wall.

#### 4.2.2    Results

Fig. 1 shows the training curves of reward and cost for our proposed methods and baselines on 2-agent and 4-agent ants walking through the corridor task. According to Fig. 1, we can find that the reward performance of both F4SG and R4SG is better than other safe MARL baselines and the performance of R4SG is better than of F4SG. Particularly, the reward performance of F4SG is close to the highest reward of unsafe HAPPO on Ant $2 \times 4$ task, and achieves the highest reward which is better than unsafe HAPPO on Ant $4 \times 2$ task. At the same time, R4SG violates the second fewer and the least safety constraints. The experiments reveal that both F4SG and R4SG quickly learn to satisfy safety constraints and achieve effective performance improvement.

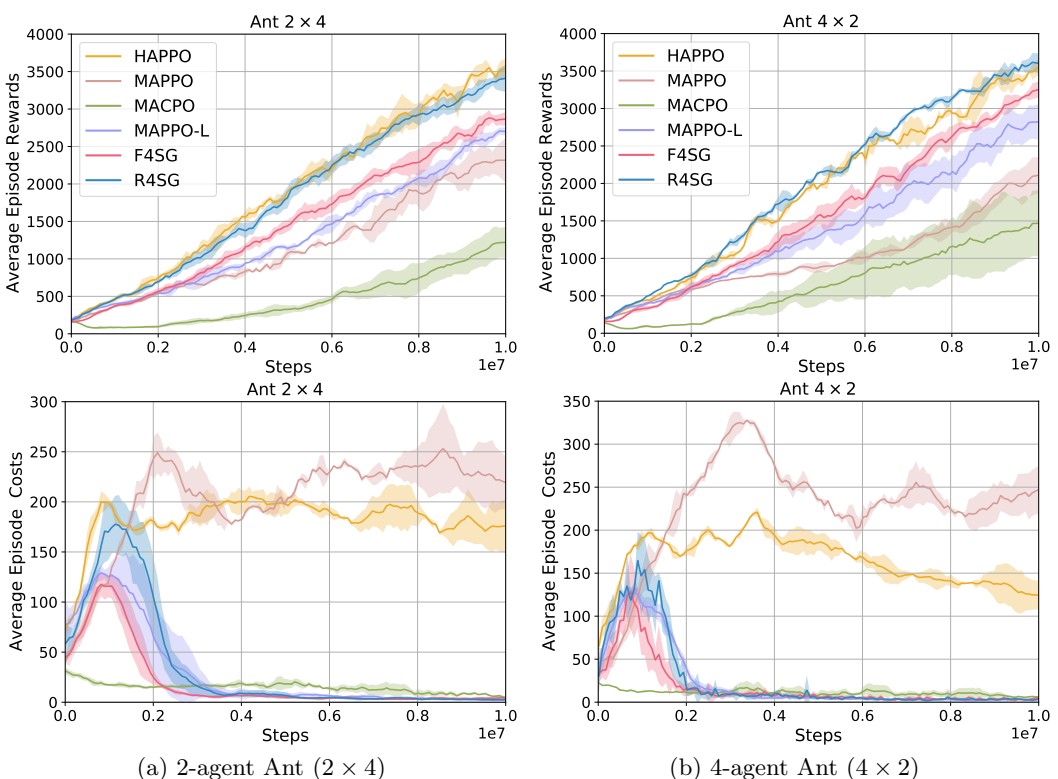

(a) 2-agent Ant $(2 \times 4)$          (b) 4-agent Ant $(4 \times 2)$

Figure 1: Performance comparisons on walking through the corridor in Safe MAMuJoCo.

### 4.3 Safe Multi-Agent Isaac Gym Environment

#### 4.3.1 Experimental Settings

Safe MAIG is developed on top of Isaac Gym (Makoviychuk et al., 2021) and provides many hands' tasks benchmark. ShadowHandOver $2 \times 6$ is a classical two-hands cooperative task. It requires two hands of fixed positions achieving the following goals: The goal of first hand with an object is finding a way to hand the item over to the second hand, and the goal of the second hand is learning how to grasp the item that is from the first hand. Hands receive a cost feedback while one finger on the first hand has safety constraints, expressed as:

$$c_t = \begin{cases} 1.0, & \text{for } |\mathbf{F}_{a4,t+1}| \geq 0.1, \\ 0, & \text{otherwise.} \end{cases}$$

where $\mathbf{F}_{a4,t+1}$ is the first hand's fourth fingers's motion degree.

#### 4.3.2 Results

Fig. 2 shows the training curves of reward and cost for our proposed methods and baselines on ShadowHandOver $2\times6$ task. According to Fig. 2, we can find that although the safe baseline MAPPO-L and F4SG both violate the lowest security constraints, F4SG achieves the highest reward performance. The reward performance of R4SG is similar to unsafe baselines MAPPO and HAPPO, as well as safe baseline MAPPO-L, and is superior to another safe baseline MACPO. Although R4SG's violation of security constraints is slightly higher than MAPPO-L, it is less than other compared baselines.

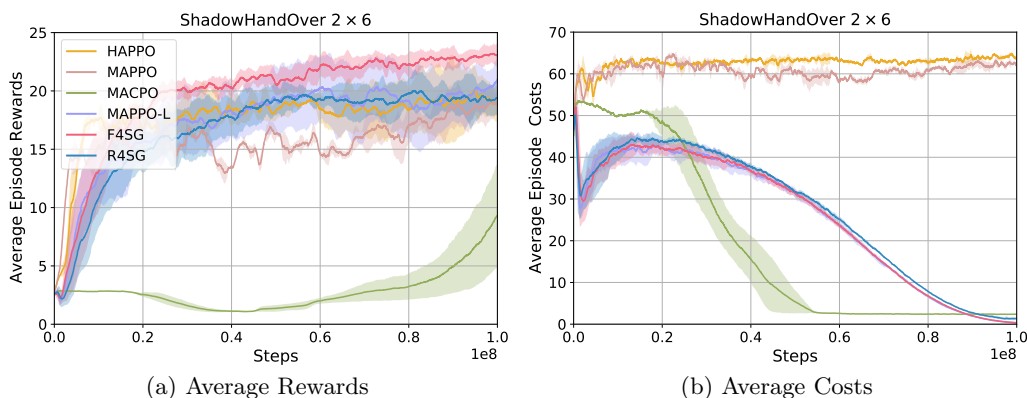

(a) Average Rewards

(b) Average Costs

Figure 2: Performance comparisons on ShadowHandOver 2×6 task in Safe MAIG.

### 4.4 Multi-Agent Power Distribution Networks Environment

#### 4.4.1 Experimental Settings

The MAPDN environment serves as a suitable framework for distributed active voltage control, readily accommodating the application of MARL methods. The 33-/141-bus network scenarios of MAPDN environment are emploied to compare the performance. In 33-bus network, there are 4 regions with 6 agents. In 141-bus network, there are 9 regions with 22 agents. The agents (photovoltaics, PVs) in the power distribution network has the potential for voltage fluctuations exceeding the power grid standards. The distributed active voltage control task (Shi et al., 2023; Qu et al., 2024) need agents to control the voltage within a safety range around a stationary value, while the reactive power generation is as less as possible. Thus, the following two key evaluation metrics are emploied to assess the performance:

- *Controllable Rate* (CR). It calculates the ratio of the time steps in each episode where all buses' voltages is under control within the safety range from 0.95 per unit to 1.05 per unit (reference voltage is 1.0 per unit).

- *Power Loss* (PL). It calculates the average of the total power loss for entire power network per time step in each episode.

#### 4.4.2 Results

Fig. 3 shows the training curves of CR and PL for our proposed methods and baselines on two network scenarios. According to Fig. 3, we can find that R4SG has the highest CR and lowest PL in both scenarios. Meanwhile, the CR and QL performances of F4SG are better than comparison baselines. This reveal that our proposed methods can adaptively scale with a larger number of agents. The performances of unsafe MARL baselines HAPPO and MAPPO in both scenarios are poor. The performances of safe baseline MAPPO-L are close to the performances of F4SG and the performances of another safe baseline MACPO are poor among the safe baselines.

### 4.5 Ablation Study

This section discusses the performance of the method combining fear and regret (Mixed) and we choose the 2 × 4 Ant walking through corridor task in Safe Multi-Agent MuJoCo environment to make the report. According to Fig. 4, we can see that the reward performance of the Mixed method is worse than that of F4SG and R4SG despite having similar cost performance. That is because considering both fear and regret makes the learned policy too pessimistic, which influences the belief of actual decision-making.

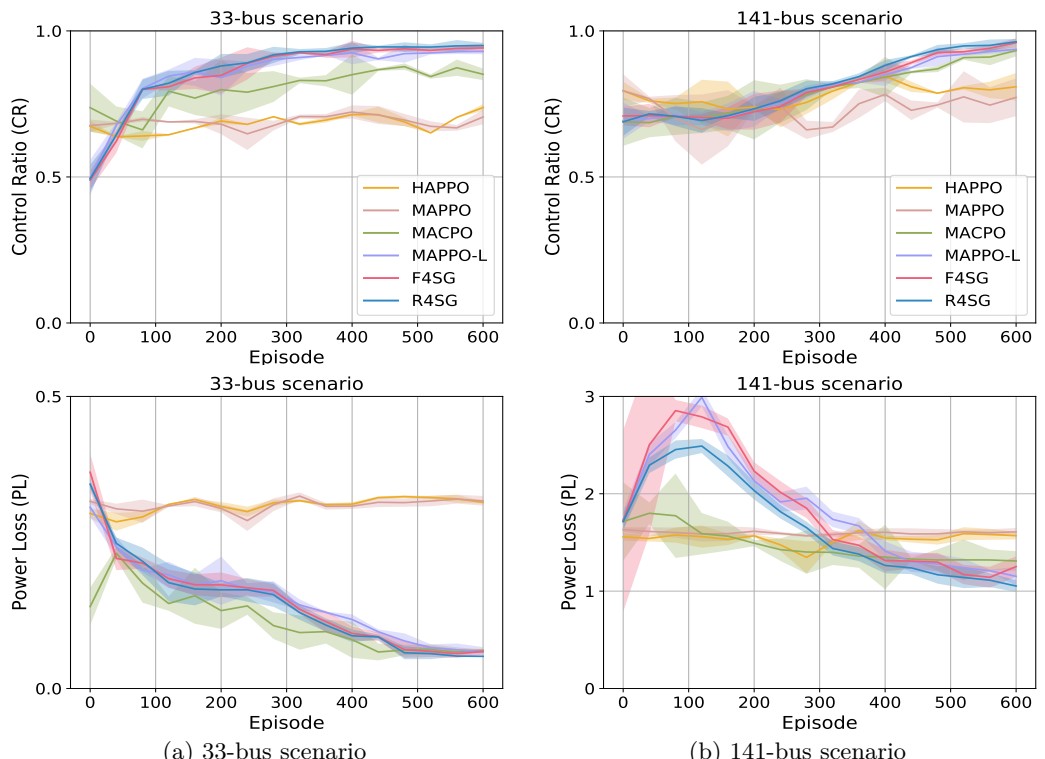

Figure 3: Performance comparisons on 33-/141-bus scenarios in the MAPDN environment.

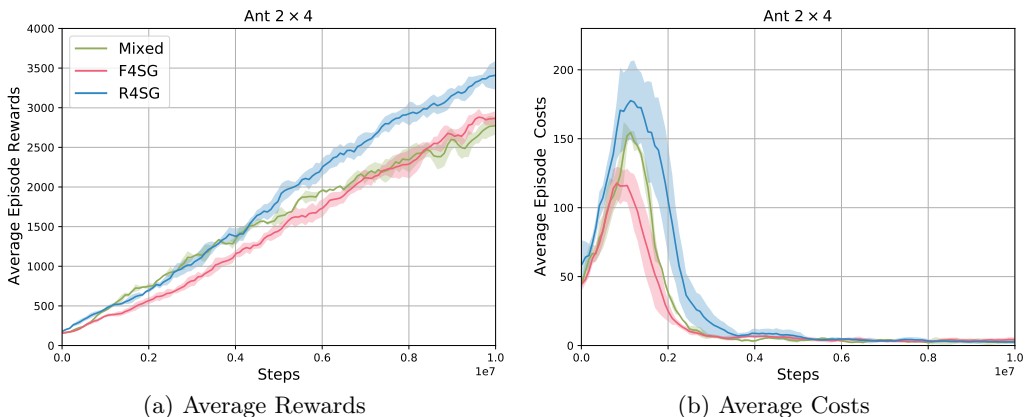

Figure 4: The Results of ablation methods on $2 \times 4$ Ant task in Safe MAMuJoCo.

## 5 Conclusion and Future Work

In this paper, we propose two novel safety guarantee methods, fear for safety guarantee (F4SG) and regret for safety guarantee (R4SG), to learn cooperative and safe strategies. These methods are inspired by the Protection Motivation Theory from social psychology, which offers a useful theoretical underpinning for guiding the learning of protective behaviors. The experimental results demonstrate the F4SG and R4SG achieve advantages in the balance between performance improvement and safety constraint satisfaction compared with state-of-the-art baselines. In the future, we will explore the possibility of extending safety MARL methods to offline problems to ensure safety implementation.

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

# A  Appendix

## A.1  Related Work

Recently, safety has been increasing concerned in the MARL domain and some safety guarantee algorithms have been studied for MARL methods.

The primal-dual framework-based safe MARL methods. Safe Dec-PG (Lu et al., 2021) is the first decentralized policy gradient algorithm for safe MARL, which leverages the min-max saddle-point formulation to find the $\epsilon$-first-order stationary points. UCB-CSAPO (Ding et al., 2023) focuses on the online safe MARL and formulates it as the generalized Lagrangian policy optimization problem, solved by the developed upper confidence reinforcement learning algorithm. The work (Gu et al., 2023) proposes two safe multi-agent policy gradient algorithms, MACPO and MAPPO-Lagrangian, for multi-robot control. MACPO leverages the multi-agent trust region method and MAPPO-Lagrangian uses Lagrangian multipliers to simplify the repetitive computation in MACPO, which achieves the goal of improving reward while satisfying safety constraints. MA-DELC (Qu et al., 2024) is a MARL method extended from the primal-dual optimization RL method, addressing the challenge of guaranteeing safety constraints in the active voltage control problem.

The shielding-based safe MARL methods. The work (Elsayed-Aly et al., 2021) develops two shielding frameworks, centralized shielding and factored shielding, to ensure multiple agents satisfy the safety specification expressed by linear temporal logic. The work (Melcer et al., 2022) discusses the strong assumptions in previous shielded MARL methods and presents a decentralized shielding algorithm for the decomposition of a centralized shield. And then, the work (Melcer et al., 2024) extends this decentralized shielding algorithm to the shielding method without any assumptions, which is applicable to environments with general partial observability. MBDS (Xiao et al., 2023) synthesizes distributive shields by an approximate world model and allows shields without prior knowledge to dynamically split, merge, and recompute based on agents' states.

The safe MARL methods with safety layer. Safe MADDPG (Sheebaelhamd et al., 2021) extends the idea of linearizing the single-step transition dynamics to the MADDPG framework and adds a safety layer to ensure safety. The work(Shi et al., 2023) proposes two MARL algorithms with a centralized data-driven safety layer, ACPL-MADDPG and ACS-MADDPG, to ensure the power system satisfies the security.

Other approach for safe MARL methods. Multi Safe Q-Agent (Zhu et al., 2020) utilizes a Gaussian Processes-based approach to estimate safety and uncertainty, enabling decentralized safe navigation for multiple different agents. SAFE-M$^3$-UCRL (Jusup et al., 2024) is the first safe model-based mean-field MARL algorithm to solve the vehicle repositioning problem, ensuring pessimistic constraints satisfaction with high probability via using epistemic uncertainty and log-barrier approach.

However, the above works ignore the great potential of utilizing human knowledge about coping with threats in guiding MARL methods to ensure safety. Thus, we leverage the theoretical underpinnings of the Protection Motivation Theory, revealing the individuals' protection behavior motivations when perceiving threats, to develop safety guarantee approaches for MARL methods.

## A.2  The Visualization of the Environments

Fig. 5 visualizes the 2- / 4- agent ant walking through the corridor task in the Safe Multi-Agent MuJoCo environment, the ShadowHandOver 2×6 task in Safe Multi-Agent Isaac Gym environment, and the 33-bus network scenario in Multi-Agent Power Distribution Networks.

## A.3  The Exact Results of Each Experiment

Table1-3 show the exact results in Safe MAMUJoCo, Safe MAIG, and MAPDN.

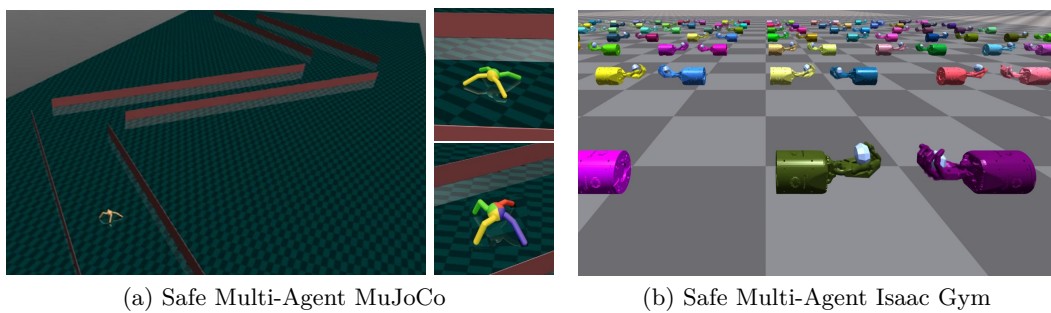

(a) Safe Multi-Agent MuJoCo        (b) Safe Multi-Agent Isaac Gym

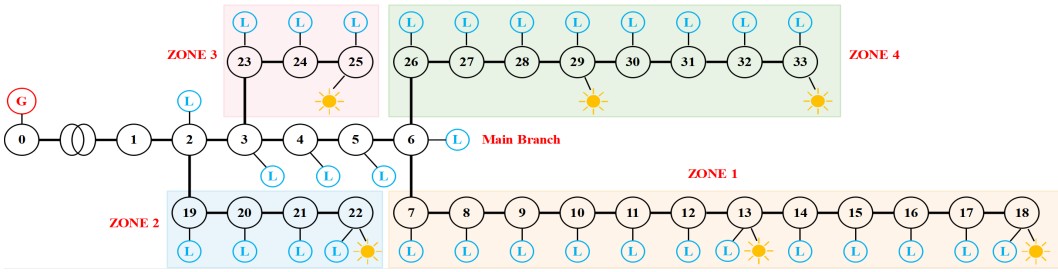

(c) Multi-Agent Power Distribution Networks

Figure 5: Visualization of multi-agent environments with safety-critical tasks. (a) The 2-agent and 4-agent ants walking though the corridor task in Safe Multi-Agent MuJoCo. (b) ShadowHandOver 2×6 task in Safe Multi-Agent Isaac Gym. (c) The 33-bus network scenario in Multi-Agent Power Distribution Networks.

Table 1: The exact results for our methods and compared baselines in Safe MAMuJoCo.

| Methods | 2-agent Ants (2 × 4) | | 4-agent Ants (4 × 2) | |
|---|---|---|---|---|
| | Average Reward ↑ | Average Cost ↓ | Average Reward ↑ | Average Cost ↓ |
| MAPPO | $2317.59 \pm 449.14$ | $219.15 \pm 26.17$ | $2109.74 \pm 240.95$ | $243.44 \pm 26.80$ |
| HAPPO | $3552.75 \pm 55.62$ | $173.61 \pm 22.17$ | $3554.51 \pm 110.54$ | $124.45 \pm 16.67$ |
| MACPO | $1220.23 \pm 206.52$ | $4.56 \pm 0.78$ | $1468.33 \pm 440.38$ | $5.88 \pm 0.04$ |
| MAPPO-L | $2706.03 \pm 46.01$ | $2.05 \pm 0.69$ | $2821.06 \pm 237.31$ | $3.21 \pm 0.70$ |
| F4SG (Our) | $2880.68 \pm 106.21$ | $4.02 \pm 1.46$ | $3254.05 \pm 106.58$ | $2.28 \pm 1.57$ |
| R4SG (Our) | $3409.12 \pm 181.29$ | $2.54 \pm 1.14$ | $3616.66 \pm 123.74$ | $2.11 \pm 0.85$ |

Table 2: The exact results for our methods and compared baselines in Safe MAIG.

| Methods | Average Reward ↑ | Average Cost ↓ |
|---|---|---|
| MAPPO | $19.01 \pm 1.44$ | $62.79 \pm 1.02$ |
| HAPPO | $20.08 \pm 2.14$ | $64.66 \pm 0.03$ |
| MACPO | $9.26 \pm 4.44$ | $2.37 \pm 0.04$ |
| MAPPO-L | $20.99 \pm 1.96$ | $0.41 \pm 0.03$ |
| F4SG (Our) | $23.21 \pm 0.89$ | $0.41 \pm 0.03$ |
| R4SG (Our) | $20.03 \pm 1.79$ | $1.99 \pm 0.18$ |

Table 3: The exact results for our methods and compared baselines in MAPDN.

| Methods | 33-bus scenario | | 141-bus scenario | |
|---|---|---|---|---|
| | Control Ratio (%) ↑ | Power Loss ↓ | Control Ratio (%) ↑ | Power Loss ↓ |
| MAPPO | $70.67 \pm 0.68$ | $0.329 \pm 0.006$ | $79.51 \pm 5.65$ | $1.63 \pm 0.02$ |
| HAPPO | $68.84 \pm 1.46$ | $0.328 \pm 0.002$ | $82.34 \pm 0.10$ | $1.62 \pm 0.10$ |
| MACPO | $86.73 \pm 0.96$ | $0.063 \pm 0.002$ | $93.28 \pm 0.88$ | $1.31 \pm 0.04$ |
| MAPPO-L | $92.98 \pm 1.01$ | $0.063 \pm 0.009$ | $93.68 \pm 2.31$ | $1.15 \pm 0.70$ |
| F4SG (Our) | $94.02 \pm 0.21$ | $0.060 \pm 0.003$ | $95.05 \pm 0.58$ | $1.14 \pm 1.57$ |
| R4SG (Our) | $94.96 \pm 0.19$ | $0.055 \pm 0.002$ | $96.28 \pm 0.47$ | $1.05 \pm 0.85$ |

### A.4 Algorithm of Fear for Safety Guarantee (F4SG)

Algorithm 1 gives an overall picture of the F4SG method. Exact details of the method can be found in the code.

---

**Algorithm 1 Fear for Safety Guarantee (F4SG)**

---

Initialize: parameters $\{\theta_0^i\}_{i \in \mathcal{N}}$ for actor networks, parameters $\{\zeta_{k,0}^i\}_{1 \le k \le m^i}^{i \in \mathcal{N}}$ for fear prior networks, parameters $\{\phi_0^i\}_{i \in \mathcal{N}}$ for critic networks, parameters $\{\phi_{k,0}^i\}_{1 \le k \le m^i}^{i \in \mathcal{N}}$ for cost networks

Set: batch size $B$, learning rates $\alpha_\theta$ and $\alpha_\lambda$, experience replay buffer $D$, number of: agents $n$, episodes $T_e$, steps per episode $T_s$, PPO epochs $E_{ppo}$.

1: **for** $\hat{t} = 0, 1, \ldots, T_e - 1$ **do**
2:      Collect a set of trajectories by running the joint policy $\boldsymbol{\pi}_{\theta_{\hat{t}}}$
3:      Collect a set of fear severity in trajectories by running the FPN
4:      Store transitions $\{(o_t^i, a_t^i, o_{t+1}^i, r_t, \tilde{f}_t^{i,k})_{t \in T_s}^{i \in \mathcal{N}, k \in m^i}\}$ into $D$
5:      Calculate advantages estimate $\hat{A}(s, \mathbf{a})$ based on critic network with GAE
6:      Calculate cost advantages estimate $\hat{A}_k^{i,(F)}(s, \mathbf{a}^i)$ for all agents and costs based on fear prior networks and cost networks with GAE    // Eq.(9)
7:      Draw a random permutation of agents $i_{1:n}$
8:      Set $L^{i_1}(s, \mathbf{a}) = \hat{A}(s, \mathbf{a})$ in Eq.(10)
9:      **for** $i_h = i_1, i_2, \ldots, i_n$ **do**
10:          Initialize a policy parameter $\theta^{i_h} = \theta_{\hat{t}}^{i_h}$ and dual variables $\lambda_k^{i_h} = 0, \forall k = 1, 2, \ldots, m^i$
11:          Sample data $d^{i_h}$ from $D$
12:          **for** $e = 1, 2, \ldots, E_{ppo}$ **do**
13:              Adam update $\theta^{i_h}$ on $\mathcal{L}_{(\lambda_k^{i_h})}^{i_h}(\theta^{i_h})$ with data $d^{i_h}$    //Eq.(10)
14:              Adam update $\zeta$ on $\mathcal{L}_k^{i_h}(\zeta)$ with data $d^{i_h}$    //Eq.(4)
15:              **for** $k = 1, 2, \ldots, m^{i_h}$ **do**
16:                  Update temporarily the dual variable $\lambda_k^{i_h}$
17:              **end for**
18:          **end for**
19:          Update the actor network parameter $\theta_{\hat{t}+1}^{i_h} = \theta^{i_h}$
20:      **end for**
21:      Adam update $\phi$ on $\mathcal{L}^C(\phi)$ and $\phi_k$ on $\mathcal{L}_k^C(\phi_k)$ //    Eq.(12a, 12b)
22: **end for**

---

### A.5 Algorithm of Regret for Safety Guarantee (R4SG)

Algorithm 2 gives an overall picture of the R4SG method. Exact details of the method can be found in the code.

### A.6 Computational Complexity

The computational complexity of both F4SG and R4SG is $\mathcal{O}(TNEMP)$, where $T$ denotes the number of steps, $N$ denotes the number of agents, $E$ denotes the number of PPO-epoch, $M$ denotes the number of constraints, $P$ denotes the max number of parameters in {actor network, and fear/regret prior network}. Due to the $P \propto |S| \times 2$ in fear prior network and the $P \propto |S| \times |A|$ in regret prior network, the wall-clock times that methods train an episode are different. For example, the average wall-clock time for training F4SG and R4SG on 2-agent ant walking through the corridor task in Safe MAMuJoCo is approximately 23.5 s/episode and approximately 25.2 s/episode respectively. Fig. 6 shows the steps of our methods and baselines that complete an episode on Ant $2 \times 4$ task.

---

**Algorithm 2 Regret for Safety Guarantee (R4SG)**

---

**Initialize:** parameters $\{\theta_0^i\}_{i\in\mathcal{N}}$ for actor networks, parameters $\{\psi_{k,0}^i\}_{1\le k\le m^i}^{i\in\mathcal{N}}$ for regret prior networks, parameters $\{\phi_0^i\}_{i\in\mathcal{N}}$ for critic networks, parameters $\{\phi_{k,0}^i\}_{1\le k\le m^i}^{i\in\mathcal{N}}$ for cost networks

**Set:** batch size $B$, learning rates $\alpha_\theta$ and $\alpha_\lambda$, experience replay buffer $D$, number of: agents $n$, episodes $T_e$, steps per episode $T_s$, PPO epochs $E_{ppo}$.

1: for $\hat{t} = 0, 1, \ldots, T_e - 1$ do
2:     Collect a set of trajectories by running the joint policy $\boldsymbol{\pi}_{\theta_{\hat{t}}}$
3:     Collect a set of regret severity in trajectories by running the RPN
4:     Process the regret severity close to human emotion using Eq.(13)
5:     Store transitions $\{(o_t^i, a_t^i, o_{t+1}^i, r_t, \tilde{r}_t^{i,k})_{t\in T_s}^{i\in\mathcal{N},k\in m^i}\}$ into $D$
6:     Calculate advantages estimate $\hat{A}(s, \mathbf{a})$ based on critic network with GAE
7:     Calculate cost advantages estimate $\hat{A}_k^i(s, \mathbf{a}^i)$ for all agents and costs based on cost networks with GAE
8:     Draw a random permutation of agents $i_{1:n}$
9:     Set $L^{i_1}(s, \mathbf{a}) = \hat{A}(s, \mathbf{a})$ in Eq.(15)
10:     for $i_h = i_1, i_2, \ldots, i_n$ do
11:         Initialize a policy parameter $\theta^{i_h} = \theta_{\hat{t}}^{i_h}$ and dual variables $\lambda_k^{i_h} = 0, \forall k = 1, 2, \ldots, m^i$
12:         Sample data $d^{i_h}$ from $D$
13:         for $e = 1, 2, \ldots, E_{ppo}$ do
14:             Adam update $\theta^{i_h}$ on $\mathcal{L}_{(\lambda_k^{i_h})}^{i_h}(\theta^{i_h})$ with data $d^{i_h}$    //Eq.(15)
15:             Adam update $\psi$ on $\mathcal{L}_k^{i_h}(\psi)$ with data $d^{i_h}$    //Eq.(14)
16:             for $k = 1, 2, \ldots, m^{i_h}$ do
17:                 Update temporarily the dual variable $\lambda_k^{i_h}$
18:             end for
19:         end for
20:         Update the actor network parameter $\theta_{\hat{t}+1}^{i_h} = \theta^{i_h}$
21:     end for
22:     Adam update $\phi$ on $\mathcal{L}^C(\phi)$ and $\phi_k$ on $\mathcal{L}_k^C(\phi_k)$ //    Eq.(12a, 12b)
23: end for

---

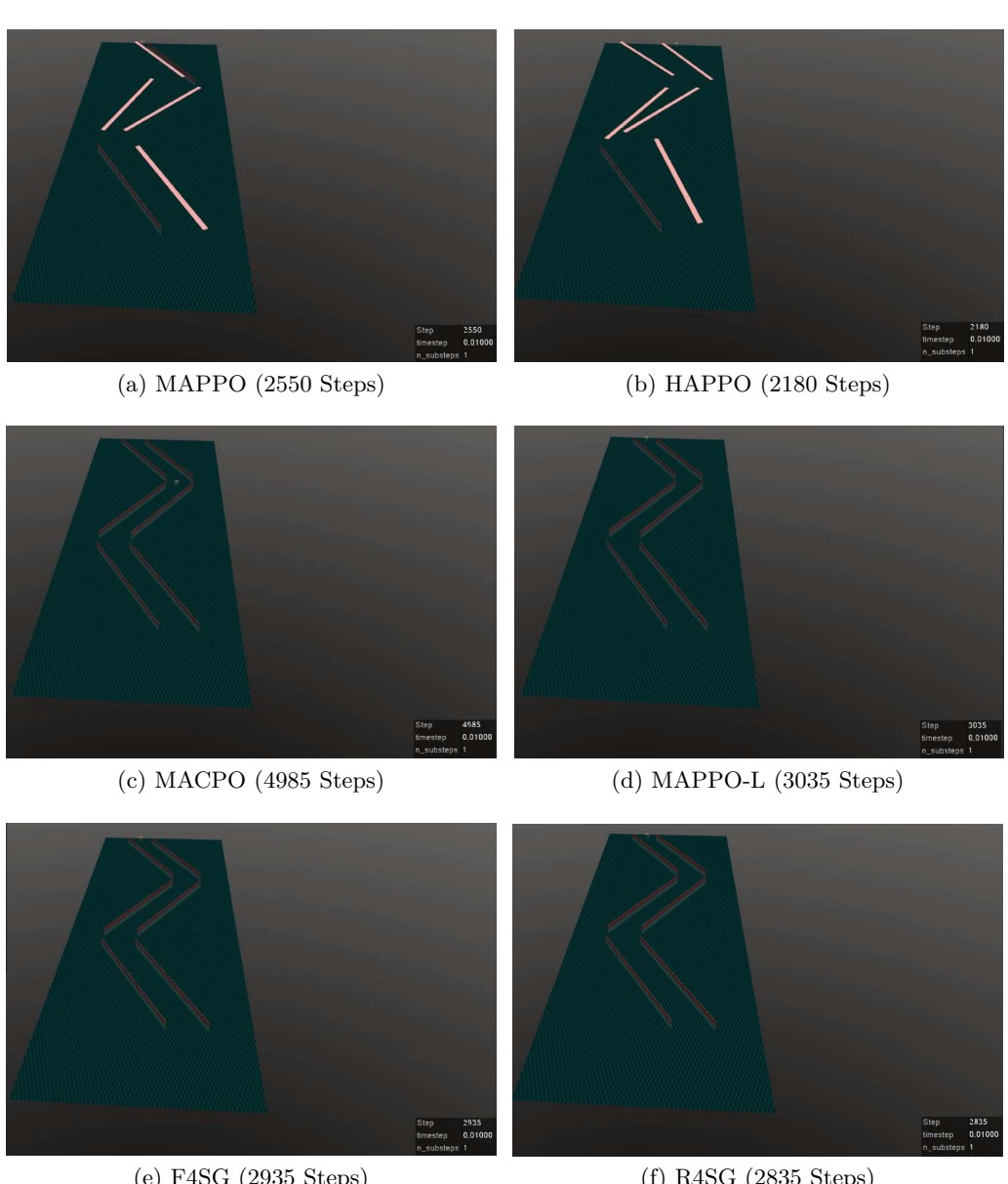

(a) MAPPO (2550 Steps)

(b) HAPPO (2180 Steps)

(c) MACPO (4985 Steps)

(d) MAPPO-L (3035 Steps)

(e) F4SG (2935 Steps)

(f) R4SG (2835 Steps)

Figure 6: Steps of our methods and baselines that complete an episode on Ant $2 \times 4$ task

