# OpenReview forum: "Safe Multi-agent Reinforcement Learning with Protection Motivation Theory"
_ICLR.cc/2025/Conference — ICLR 2025 Conference Withdrawn Submission_

### Official Review · Reviewer_kPnq · 2024-10-22

**Soundness:** 2
**Presentation:** 2
**Contribution:** 1
**Rating:** 3
**Confidence:** 4

**Summary:**

The paper leverage protection motivation theory (PMT), a phenomenon in human risk assessment for safe MARL. The method for safe MARL mimics PMT by modelling fear and regret to assess threat severity and learn protective behaviors by not visiting certain states or taking certain actions. The algorithms that model fear and regret is called F4SG and R4SG. Experiment result demonstrates that their proposed method are safer and more efficient than state-of-the-art safe MARL algorithms.

**Strengths:**

The paper studies an important problem of safety in MARL. It is clearly written and motivated by human behavior.

**Weaknesses:**

1. The motivation seems unclear to me. The author use human behavior as motivation, but failed to point out what problems exists in current works, as been discussed in Introduction section. If incorporating human behavior is a must, then it should be solving some limitations of current works, yet this part is missing.

2. The authors should consider evaluating their method on more tasks. Safe-MAMujoco, Safe MAIG and MAPDN all contains 10-20 tasks, yet the authors evaluated only 2 tasks in Safe-MAMujoco and 1 in Safe MAIG. The authors can consider adding additional 3-6 tasks on Safe-MAMujoco and Safe MAIG.

3. The gain in safety seems minor in Fig. 1, 2 and 3, especially comparing with MACPO. I would say there is a strong overlap between the curves of proposed method and MACPO. I would suggest authors to evaluate the safety measure on some more challenging tasks.

4. The problem of safe MARL is not a MDP. Typically, MDP modells the decision process of single-agent, when in multi-agent case, it's commonly formulated as a Dec-POMDP or Markov Game. So the problem formulation is incorrect. According to experiments I guess it's some sort of safe Dec-POMDP. Also refer to MACPO for their problem formulation.

5. The author should add surveys on safe MARL literature in preliminaries.

6. In Sec. 3.1.2, many derivations are based on existing literatures. Maybe it is better to focus on the central derivations.

7. What are the guarantees for "fear for safety guarantee"? I suppose it to be some type of bounds, but failed to find any.

Minor: Seems the paper do not follow the ICLR template and exceeds the page limit. Also, there are many grammar errors (eg, In this paper, we introduce PMT into the MARL to address the challenge safety. in line 067-068).

**Questions:**

See "Weakness" section.

---

### Official Review · Reviewer_W8Xp · 2024-10-29

**Soundness:** 3
**Presentation:** 3
**Contribution:** 2
**Rating:** 5
**Confidence:** 4

**Summary:**

The authors propose two algorithms F4SG and R4SG to enhance agents’ safety in reinforcement learning. F4SG and R4SG are designed with the concepts in protection motivation theory. Fear and Regret are learned to provide safety in two algorithms, respectively. Then agents are optimized with Lagrange dual and trust region. Experiments are conducted on three different tasks.

**Strengths:**

1. The authors summary many related works of safe MARL.
2. The story from protection motivation theory makes the proposed algorithms more intuitive.
3. Experiments in three different tasks are conducted with 2 MARL baselines and 2 safe MARL baselines.

**Weaknesses:**

1. Some components of the algorithms are not clear, especially for the optimization of FPN and RPN.
2. The application of Lagrange dual is a main component of the proposed algorithms, while it has been used in many related works. Besides, the learning of FPN and RPN is more like the learning of cost function.
3. In the experiments, it seems that curves have not converges, or the performance of proposed algorithms is not obviously better than baselines.

**Questions:**

1. Could the authors explain equation 3 more clearly?
What are the dimensions of fi (FPN)? What’s the meaning of its different index, which is not clear enough in line169.
How is Sd chosen?
Does the first term of equation 3 mean the learning of cost function? This idea is used in many prior works, such as “Safe Reinforcement Learning Using Advantage-Based Intervention”.
2. Similarly, the authors are expected to explain equation 14 more clearly. Are fear and regret only applicable for discrete action space?
3. Is there anything novel in 3.1.2 and 3.2.2, except the use of Fear and Regret, in comparison with prior works using Lagrange dual?
4. It seems that for each episode in F4SG and R4SG, parameters are updated E_ppo times. What’s the update frequency of baseline algorithms? Is it the reason why F4SG and R4SG converge faster than baselines in 4.2?
In 4.3 and 4.4, it seems that MAPPO-L achieves similar performance as F4SG and R4SG when they all converge

---

### Official Review · Reviewer_t71B · 2024-10-30

**Soundness:** 3
**Presentation:** 3
**Contribution:** 3
**Rating:** 5
**Confidence:** 3

**Summary:**

This paper aims to enhance safety in multi-agent reinforcement learning (MARL) by integrating "fear" and "regret" emotions, inspired by Protection Motivation Theory (PMT). Two methods are introduced: Fear for Safety Guarantee (F4SG) and Regret for Safety Guarantee (R4SG), which evaluate threat severity in states and actions to help agents avoid unsafe states or actions. Experimental results demonstrate that F4SG and R4SG effectively reduce safety violations in multi-agent environments while achieving high performance under safety constraints.

**Strengths:**

(1)	This paper attempts to introduce emotion modeling into multi-agent reinforcement learning, employing fear and regret to adjust agent’s decision-making behaviors. This interdisciplinary innovation brings a compelling perspective to the study.

(2)	This paper provides a detailed theoretical modeling of the proposed F4SG and R4SG methods and establishes a solid theoretical foundation for emotion modeling through mathematical formulations.

(3)	The experimental section demonstrates the performance of F4SG and R4SG across different task scenarios, indicating that emotion modeling can achieve high performance while ensuring the safety of agents.

**Weaknesses:**

(1)	This paper introduces “fear” and “regret” for pre-decision risk assessment and post-decision reflection respectively. However, the mixed model doesn’t enhance performance, which contradicts real-world scenarios where humans often experience multiple emotions simultaneously. An effective framework to integrate the two emotions is lacking.

(2)	The experimental analysis is relatively brief. Since the paper proposes two emotion models, it should provide a more detailed comparative analysis of their effectiveness in different scenarios and explore suitable application contexts to better guide practical use of the methods.

(3)	This paper lacks a time complexity analysis, which limits the evaluation of the model’s feasibility for real-world use.

**Questions:**

(1)	The motivation part (page 2, lines 58-66) mentions that PMT includes multiple emotions; why were only fear and regret selected for modeling in this study?

(2)	In the optimization of the Fear Prior Network (FPN), the quantification of fear severity relies on a prior distribution (line 137). Could this lead to instability in new or uncertain environments?

(3)	Fear and regret are emotions that can naturally coexist. However, the ablation study shows that the combined model does not yield better results (page 9, lines 481-485), with the authors suggesting that it leads to overly conservative behavior. Has any exploration been done on developing a framework that effectively integrates these two emotions?

(4)	The authors propose two separate emotion models without integration and only describe the experimental results without analyzing why each emotion adapts to different scenarios (pages 7-9, results part). Could you add an analysis in the experimental section on this aspect? Otherwise, the paper merely presents two methods without a deeper exploration of their contextual suitability.

---

### Official Review · Reviewer_4qE1 · 2024-11-02

**Soundness:** 2
**Presentation:** 1
**Contribution:** 2
**Rating:** 3
**Confidence:** 2

**Summary:**

The paper proposes two safety assurance methods, fear for safety guarantee (F4SG) and regret for safety guarantee (R4SG), for cooperative and safe strategies in multi-agent systems. Drawing on the Protection Motivation Theory from social psychology, the authors provide a theoretical framework to guide the development of protective behaviors in learning agents. Experimental results show that these methods achieve a promising balance between performance gains and adherence to safety constraints, showing advantages over existing state-of-the-art approaches.

**Strengths:**

The paper is inspired by  Protection Motivation Theory and proposed two safety assurance methods, the perspective seems novel. The experimental results shows the methods are effective.

**Weaknesses:**

I find the paper difficult to follow; many equations are listed without interpretations. Additionally, the paper lacks a comprehensive discussion of related work. While PMT serves as good inspiration for the method, I am not entirely sure how the essence of the proposed methods differs from other traditional safe MARL methods.

**Questions:**

1. Could you provide detailed interpretatiosn for equations

2. Could you add discussion of related works?

3. Except the perspective inspired by PMT, could you discuss the novelty of your methods? How do your methods differ from other traditional methods?

---

### Official Review · Reviewer_nnso · 2024-11-04

**Soundness:** 3
**Presentation:** 1
**Contribution:** 2
**Rating:** 3
**Confidence:** 4

**Summary:**

This paper developed a Safe Multi-agent Reinforcement Learning Method based on the Protection Motivation Theory (PMT). The authors proposed to utilize two emotional mechanisms, fear and regret, to design fear for safety guarantee (F4SG) and regret for safety guarantee (R4SG) to improve the current primal-dual safe MARL pipeline. Experiments on safe MARL benchmarks validate the security and efficiency of their algorithms compared with SOTA baselines.

**Strengths:**

The idea of ​​applying PMT to the Safe MARL pipeline seems quite novel, and extensive experiments on the Safe MARL benchmark validate the superiority of the proposed approach in further minimizing the cumulative cost.

**Weaknesses:**

However, some weakness significantly hinders readers from further evaluating the contribution and importance of the work:

1.	Annotation & Mathematical Derivation: the presentation of the work, especially regarding the theoretical part (part 3), is very chaotic. First, many annotations are not introduced during mathematical derivations. For example, in your introduction of FPN, $\tilde{f}^i=F P N\left(s; \zeta^i\right)$, what is $\zeta$ here? Also, in Equation (10), what are $B$ and $T_s$ here? Each annotation should be introduced when it first appears in the paper.

2.	Proposed Theoretical and Loss Function Design: I do agree introducing the fear and regret mechanism is interesting, but why should the loss function of your FPN and RPN have loss functions like Equation (4) and (14)? What is the theoretical intuition and explanation for Equation (4) and (14)? Also, in Equation (3), why does the cost function suddenly have probability distribution $p(C^i)$? In Equation (13), what does the cost function $\mathcal{C}\left(s, a^i\right)=1$ and $\mathcal{C}\left(s, a^i\right)=0$ mean?

3.	Experiments and Hyperparameters: The experimental section needs more details about the hyperparameters used in your network training - what are the specific hyperparameter settings for each algorithm, including yours? Also, while you show the average costs, what's the actual constraint violation rate for each method? Additionally, I see you focus on improving the Lagrangian safe RL approach, but how does your method compare with those algorithms that claim zero constraint violation, like [1]?

4. The proposed PMT framework doesn't seem specifically designed for multi-agent settings - would it work equally well in single-agent scenarios? What's your motivation for choosing a multi-agent setting? The paper needs to better justify why the PMT framework is particularly suitable or important for multi-agent rather than single-agent problems.

[1] Liu T, Zhou R, Kalathil D, et al. Learning policies with zero or bounded constraint violation for constrained mdps[J]. Advances in Neural Information Processing Systems, 2021, 34: 17183-17193.

**Questions:**

Please check the weakness section.

---

### Note · Authors · 2024-11-13

I have read and agree with the venue's withdrawal policy on behalf of myself and my co-authors.